# STREAMMEM: QUERY-AGNOSTIC KV CACHE MEMORY FOR STREAMING VIDEO UNDERSTANDING

## ABSTRACT

Multimodal large language models (MLLMs) have made significant progress in visual-language reasoning, but their ability to efficiently handle long videos remains limited. Despite recent advances in long-context MLLMs, storing and attending to the key-value (KV) cache for long visual contexts incurs substantial memory and computational overhead. Existing visual compression methods require either encoding the entire visual context before compression or having access to the questions in advance, which is impractical for long video understanding and multi-turn conversational settings. In this work, we propose **StreamMem**, a *query-agnostic* KV cache memory mechanism for streaming video understanding. Specifically, StreamMem encodes new video frames in a streaming manner, compressing the KV cache using attention scores between visual tokens and generic query tokens, while maintaining a fixed-size KV memory to enable efficient question answering (QA) in memory-constrained, long-video scenarios. Evaluation on three long video understanding and two streaming video question answering benchmarks shows that StreamMem achieves state-of-the-art performance in query-agnostic KV cache compression and is competitive with query-aware compression approaches.

## 1 INTRODUCTION

Recent advances in Multimodal Large Language Models (MLLMs) (Hurst et al., 2024; Zhang et al., 2024g; Comanici et al., 2025; Bai et al., 2025; Zhang et al., 2025a) enable the capability to reason across textual and visual contents. Despite fast improvements, their capabilities to capture fine-grained details of actions, motions, object locations, interactions between objects, and spatial-temporal orders of events in long videos are still limited (Zhang et al., 2025c). There are two main reasons for this. Firstly, encoding the frames in a long video often generates a large number of visual tokens, exceeding the context length of the underlying Large Language Model (LLM). Secondly, storing the KV cache of these large number of visual tokens and attending to them during decoding poses significant memory and computational overhead. While the first issue has been alleviated by recent progress in long-context LLMs (Xiong et al., 2023; Su et al., 2024; Zhang et al., 2024b), the memory and compute efficiency of dealing with long videos remains a challenge, especially for real-world applications on edge devices.

A number of recent works explored video token compression strategies to tackle long video understanding, including temporal compression (Tang et al., 2025; Tan et al., 2024), spatial compression (Chen et al., 2024a; Zhang et al., 2025b), and hybrid methods (Zhang et al., 2024d; He et al., 2024; Shen et al., 2024; Tao et al., 2025). These approaches can suffer significant information loss. For example, the action information in the video often cannot be captured with any single frame in the video. Many such methods also rely on having access to the text query for visual compression (Li et al., 2024b; Liang et al., 2024; Hu et al., 2025), which is often unknown at the time of video processing in real-world applications (Kim et al., 2025b).

In parallel to these efforts, recent works start to explore streaming video processing with MLLMs, a paradigm in which video frames are incrementally encoded as they arrive, without prior knowledge of the video's full length or the downstream query. Compared to offline video processing, the streaming video processing setup is much more flexible, as the model does not need to know the text query or the length of the video when encoding visual information. ReKV Di et al. (2025) is

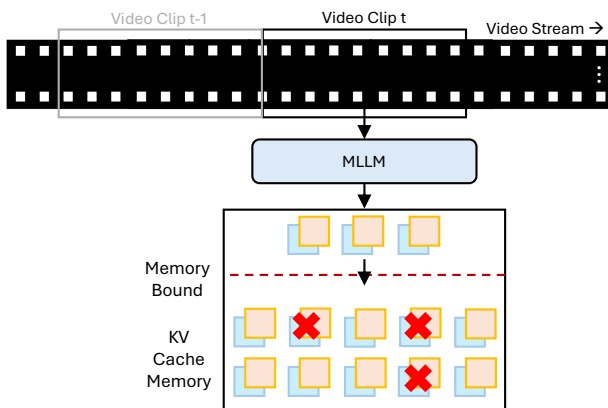

Figure 1: **Query-agnostic key-value (KV) cache compression in streaming video.** StreamMem addresses the challenge of streaming video processing under a memory budget by introducing a query-agnostic KV compression strategy.

a leading work in this direction. It encodes new video frames in the stream with sliding window attention and stores the KV cache. When the model receives a question, it retrieves the most relevant KV cache in each layer with in-context retrieval. While this method is shown to be effective, it consumes significant memory to store all the KV cache. Offloading the KV cache to memory or disk and reloading them upon retrieval could also be very inefficient as the video becomes longer. LiveVLM Ning et al. (2025) proposes a KV compression mechanism to reduce the KV cache size by 70%. While LiveVLM alleviates the issue of memory consumption, it simply throws out the KV cache of earlier tokens when the memory upper bound is reached, which can lead to complete forgetting of earlier parts of the video.

To enable efficient long video processing in memory-constrained environments, we introduce StreamMem, a training-free and query-agnostic KV cache memory system for streaming video understanding with MLLMs. StreamMem maintains a bounded memory footprint by continuously compressing the KV cache after each incoming video clip, thus preventing out-of-memory (OOM) errors and avoiding costly memory offloading regardless of video length. To achieve effective and efficient memory retention, StreamMem leverages a novel saliency metric based on cross-attention scores between visual tokens and *chat template* tokens, allowing it to select and preserve informative visual content in a query-agnostic manner. In addition, it incorporates an input frame compression module to reduce frame-level redundancy prior to MLLM encoding, and a frame-wise KV merging mechanism that constructs prototype representations for each observed frame. Together, these components produce a diverse yet compact KV cache that supports accurate and memory-efficient streaming question answering.

We evaluate StreamMem across three offline and two streaming long video understanding benchmarks (EgoSchema (Mangalam et al., 2023), MLVU (Zhou et al., 2025), VideoMME (Fu et al., 2025); RVS-Ego and RVS-Movie (Zhang et al., 2024a)) using three open-source pre-trained MLLMs (LLaVA-OneVision (Li et al., 2024a), Qwen2-VL (Wang et al., 2024a), and Qwen2.5-VL (Bai et al., 2025)). Results show that StreamMem consistently retains high utility while keeping the KV cache compact across videos of varying lengths and question types. It not only surpasses state-of-the-art streaming video models, but also achieves competitive performance with methods that rely on significantly larger memory budgets. Comprehensive ablation studies confirm the contribution of each component in the StreamMem framework. By enabling continuous, scalable memory compression without fine-tuning, StreamMem provides a crucial step toward building real-time MLLM agents capable of continuous video understanding in open-world settings.

## 2 RELATED WORK

**Streaming video understanding with MLLMs.** Streaming video understanding refers to the setting where the model continuously processes video frames in real-time. The model does not know

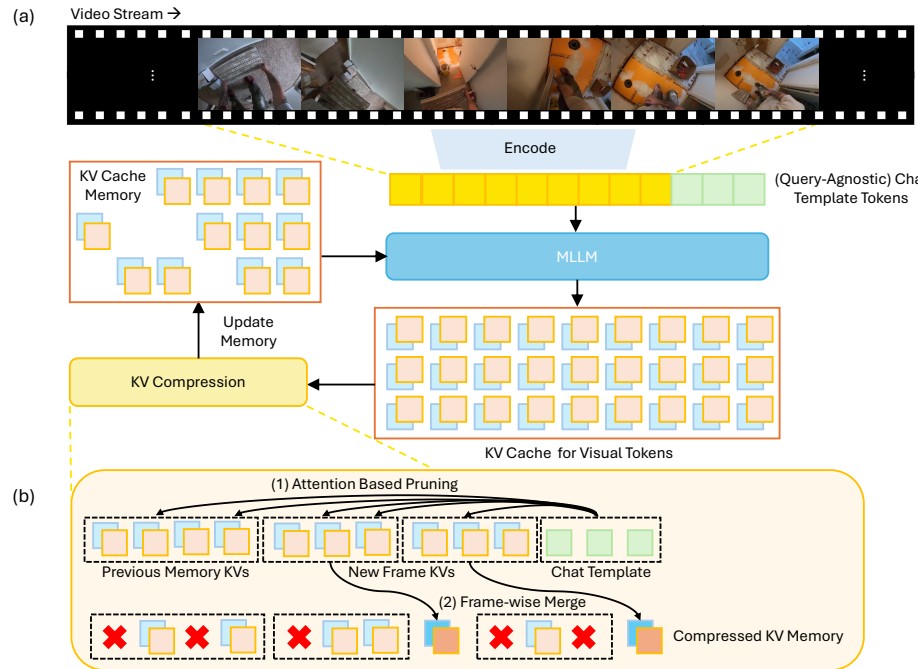

Figure 2: (a) The overall workflow of StreamMem for streaming video understanding. Incoming frames are first filtered to reduce redundancy, then passed through the vision encoder and integrated with the existing KV memory via cross-attention. The resulting KV cache is compressed to maintain a fixed memory budget, enabling continual processing of future frames or downstream question answering. (b) Detailed illustration of the KV compression module. Some KV cache in the memory and the new frames are pruned according to the attention score between the keys and the proxy queries. In addition, we aggregated the key-value pairs for each new frame into a single frame-level prototype via weighted merging (shown in darker squares). This combination of pruning and merging ensures compact yet expressive memory representations for long video sequences.

the length of the video beforehand and therefore cannot sample a fixed number of frames uniformly from the video. VideoLLM-online (Chen et al., 2024a) presents an MLLM that supports efficient streaming video processing and real-time dialogues. However, it aggressively down-samples each video frame to only include 10 visual tokens, limiting its understanding of fine-grained details in the video. Flash-VStream (Zhang et al., 2024a) and Dispider (Qian et al., 2025) use external memory modules to compress and organize visual tokens. Upon receiving a question, the model retrieves relevant visual tokens, combines them with the text tokens, and feeds them through the MLLM. Recent works start to explore KV cache compression and retrieval for video understanding. ReKV (Di et al., 2025) encodes the video in streaming fashion and stores all the KV cache by offloading to memory or disk, and performs in-context retrieval of the relevant KV cache for each layer when answering a question. The offloading of KV cache could incur a lot of memory and is not scalable to ultra-long videos. LiveVLM (Ning et al., 2025) designs a KV cache compression strategy for MLLMs to significantly reduce memory usage and improve question answering speed compared to ReKV. However, it uses a fixed compression ratio throughout the video and relies on first-in-first-out (FIFO) strategy to maintain a constrained memory, which leads to forgetting of earlier information in long videos, even though they might be informative. StreamMem resolves this issue by compressing the KV cache memory and the KVs from the new frames together and ensures a fixed-size KV cache memory throughout the video stream. Concurrent work InfiniPot-V Kim et al. (2025b) also studies streaming video processing with constrained memory consumption. Different from StreamMem, they used a combination of two compression mechanisms, temporal-axis redundancy reduction and value norm-based selection.

**Long video understanding with MLLMs.** Long video understanding has been a great challenge for MLLMs given their constrained context length. Early models such as LLaVA (Liu et al., 2023;

2024) can only process a very small number of frames, leading to significant information loss. A number of training-based methods Zhang et al. (2024b); Shen et al. (2024); Shu et al. (2025); Liu et al. (2025a) have been proposed to reduce the number of visual tokens needed to represent each video frame. In addition, recent foundation models like Gemini (Comanici et al., 2025) and Qwen2.5-VL Bai et al. (2025) also have inherent long visual context processing capability. These training-based methods, however, are often very computationally expensive, especially when fine-tuning large MLLMs, and needs re-training for new foundation models. Training-free long video understanding methods (Wang et al., 2024b; 2025; Liu et al., 2025b; Zhang et al., 2024d) compress the input visual tokens or the KV cache without the need to fine-tune the model, providing more flexibility for plug-and-play usage in new and more powerful MLLMs. StreamMem draws inspirations from the training-free methods for KV cache compression of video tokens, but focuses on the streaming setting where neither the length of the video nor the query is known during memory-constrained video encoding.

**KV cache compression in LLMs.** KV cache compression methods aim to greatly improve both memory and time efficiency of LLMs when operated in long input contexts. A number of methods explored leveraging the cross-attention weights between the query and the context to identify the most important entries in the KV cache for LLMs (Zhang et al., 2023; Li et al., 2024c; Xu et al., 2024; Fu et al., 2024). This strategy is also adopted in MLLMs for efficient visual understanding (Chen et al., 2024b; Zhang et al., 2024f). However, the query might not be available when the model processes the long context in many real-world scenarios, limiting the applicability of the query-dependent approach. To eliminate this dependency, some recent works explored query-agnostic KV cache compression mechanisms (Ge et al., 2023; Devoto et al., 2024; Hooper et al., 2024; Park et al., 2025; Kim et al., 2025a). Similar to this work, Zhang et al. (2024c) and Arif et al. (2025) explored using the attention weights of the [CLS] token for KV cache compression in MLLMs. In between query-dependent and query-agnostic methods, there are also methods which use task instructions or task-specific proxy prompts (Kim et al., 2024; Corallo et al., 2025). Stream-Mem belongs to the most flexible category of query-agnostic methods and does not need full-context encoding, making it suitable for streaming encoding of long videos.

## 3 PRELIMINARIES

**Offline video understanding with MLLMs.** The standard approach to offline video understanding with MLLMs proceeds as follows. Given a long video, a fixed number of frames $f_1, ..., f_T$ are uniformly sampled from the video, where $T$ is determined based on the model's context length or computational and memory constraints. The frames are then passed through the model's vision encoder (typically comprising a Vision Transformer (ViT) backbone (Dosovitskiy et al., 2021; Zhai et al., 2023; Zhang et al., 2024e) and a projection layer) to get $N$ visual tokens. The visual tokens are concatenated with the text tokens, including system prompts (preceding the visual tokens) and user queries (following the visual tokens), and the entire sequence is fed into the LLM. The LLM then generates a response via autoregressive decoding. To accelerate decoding, key-value (KV) caches are constructed during this process.

**KV cache compression for MLLMs in streaming video.** In streaming video processing with MLLMs, the video length is typically unknown in advance, precluding uniform frame sampling strategies used in the offline settings. At each time step $t$, the model receives a new video clip $v_t$ (a fixed-length frame segment), encodes it into a sequence of visual tokens, and forwards them through the LLM. The model then generates the corresponding key and value matrices $K_t^i$ and $V_t^i$ at each transformer layer $i$, by attending to all accumulated visual tokens from prior clips.

However, naively storing all keys and values over time leads to linear growth in memory, which is infeasible for long videos. This motivates the need for KV cache compression mechanisms that maintain a fixed memory footprint. We denote the compressed key and value matrices at time step $t$ and layer $i$ as $K_t^{i'}$ and $V_t^{i'}$, respectively. The objective is to compute compressed representations:

$$K_t^{i'}, V_t^{i'} = \text{Compress}(K_{t-1}^{i'}, K_t^i, V_{t-1}^{i'}, V_t^i),$$

subject to the global memory constraint:

$$\sum_{i=1}^{L} \|K_t^{i'}\|_0 \leq M,$$

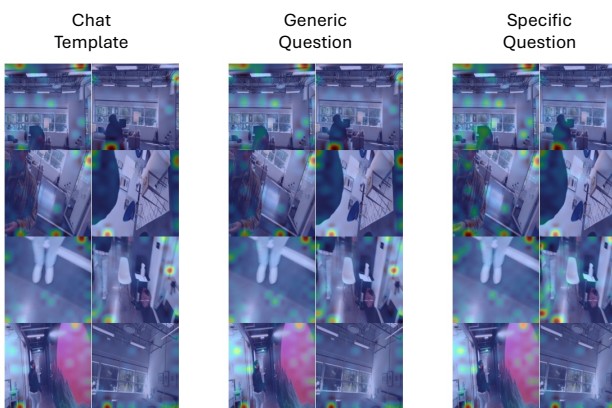

Figure 3: **Visualization of visual tokens attended to by different text queries.** Red indicates higher attention scores. Despite minor variations, different text queries attend to largely overlapping regions of the input images. The "Generic Question" is *"What is happening in the video?"*, while the "Specific Question" is *"What occurs just before reading the magazines?"* Attention scores are averaged across all layers and heads, and then interpolated from 14×14 to 384×384 to match the image resolution. The MLLM used is LLaVA-OneVision, and the video clip is sourced from the RVS-Ego benchmark (which uses videos from the Ego4D dataset (Grauman et al., 2022)).

where $L$ is the number of transformer layers in the MLLM and $M$ is the total memory budget for all layers combined.

The design of effective compression strategies that retain essential temporal information while bounding memory usage is a key challenge in streaming long video processing with MLLMs. Existing approaches such as ReKV (Di et al., 2025) and LiveVLM (Ning et al., 2025) do not address this constraint effectively, as their KV cache grows linearly over time, resulting in unbounded memory consumption for long videos.

## 4 METHOD

We now describe the key components of StreamMem, our proposed framework for efficient streaming video understanding with MLLMs. At each time step $t$, a new segment of frames is received from the video stream. These frames first undergo an input filtering step to remove temporal redundancy. The filtered frames are then encoded by the vision encoder and processed by the MLLM to produce key-value (KV) representations $\{K_t^i, V_t^i\}_{i=1}^L$ at each transformer layer $i$.

To prevent unbounded memory growth over time, the newly computed KVs are merged with the compressed KV memory from the previous time step, $\{K_{t-1}^{i\prime}, V_{t-1}^{i\prime}\}_{i=1}^L$, and passed through a compression module. This module applies two complementary strategies: (1) a novel attention-based pruning method that leverages cross-attention scores between proxy query tokens and visual tokens, and (2) a frame-wise KV merging mechanism that condenses spatial information into compact prototype representations. The output of the compression module forms the updated memory $\{K_t^{i\prime}, V_t^{i\prime}\}_{i=1}^L$, which is used by the MLLM at the next time step. An overview of the full pipeline is illustrated in Figure 2, and the KV compression procedure is detailed in Algorithm 1.

### 4.1 INPUT FRAME FILTERING

Before processing by the MLLM, each incoming video clip (a chunk of consecutive frames) is passed through a lightweight filtering step to reduce temporal redundancy. Given a sequence of frames, we compute their visual embeddings using the vision encoder. For each consecutive pair of frames, we measure the cosine similarity between their embeddings. If the similarity exceeds a predefined threshold $\delta$, the two frames are deemed redundant and their representations are merged by simple averaging.

This lightweight filtering step is similar to the temporal compression used in LongVU (Shen et al., 2024). In contrast to previous streaming approaches that rely on sliding window attention (Di et al., 2025; Ning et al., 2025), our method explicitly reduces redundancy in the input space. This ensures that highly similar frames (common in static scenes or high-frame-rate videos) do not overwhelm the KV cache with repetitive information, ultimately preserving the diversity and informativeness of the stored memory.

## 4.2 KV Cache Memory

After frame-wise token compression, the retained visual tokens of the current video segment $v_t$ are concatenated with a set of auxiliary query tokens and passed into the MLLM. The model computes key-value pairs $\{K_t^i, V_t^i\}_{i=1}^L$ in each transformer layer $i$, attending over both the current tokens, the tokens in the previous time step, and the compressed KV cache from previous time step, $\{K_{t-1}^{i\prime}, V_{t-1}^{i\prime}\}_{i=1}^L$.

To guide the KV cache compression process, we rely on the cross-attention scores between the auxiliary query tokens and the visual tokens. This attention-based saliency measure has proven effective in prior works (Chen et al., 2024b; Wang et al., 2025) for real user queries. However, unlike those settings, our method operates under a query-agnostic streaming setup, where the user query is unavailable at the time of visual token selection.

To approximate a generic query, we leverage the system's chat template tokens as a proxy. Specifically, we use the tokens: `<|im_end|><|im_start|>assistant\n`, which we append after the visual tokens. Due to the prevalence of video captioning data during the MLLM pretraining, this implicitly prompts the MLLM to generate a generic video description even in the absence of an explicit question. As a result, we expect the model to implicitly attend to informative visual content in this setup.

Formally, let $Q \in \mathbb{R}^{q \times d}$ be the query representation of the chat template tokens at a given layer, and $K_t$ be the key matrices of the visual tokens from the KV memory and the current clip. The cross-attention scores are computed as:

$$A_t^i = \mathrm{Softmax}\left(\frac{Q(K_t)^\top}{\sqrt{d}}\right), \tag{1}$$

where $A_t^i$ denotes the attention weights from chat template tokens to visual tokens. We aggregate these scores (e.g., by averaging over $q$) to obtain an importance score for each visual token, which we use to select the top-$k$ most salient visual tokens to retain in the compressed cache from each layer. The memory budget is even distributed across all layers.

In addition to pruning, StreamMem further compresses memory via KV merging. Inspired by frame-level merging in MLLMs (He et al., 2024) and visual token merging (Zhang et al., 2024d), we compute a *prototype* key and value representation for each frame. This is done by computing a weighted average of the keys and values based on the normalized attention scores:

$$\bar{K}_t^i = \sum_{j=1}^n \alpha_j^i \cdot K_{t,j}^i, \quad \bar{V}_t^i = \sum_{j=1}^n \alpha_j^i \cdot V_{t,j}^i, \tag{2}$$

where $\alpha_j^i$ denotes the normalized importance score of the $j$-th visual token at layer $i$.

These prototype representations $\bar{K}_t^i, \bar{V}_t^i$ are inserted at the end of the selected token sequence from $v_t$, preserving frame-wise temporal alignment via position IDs. Therefore, the final compressed cache $\{K_t^{i\prime}, V_t^{i\prime}\}$ consists of a mix of salient visual tokens and frame prototypes, enabling both fine-grained and global memory retention.

## 4.3 Positional Embedding

MLLMs are typically not extensively trained on long video sequences due to the scarcity of high-quality, long-form video-text data. As a result, despite the long context lengths supported by the underlying language models, MLLMs often struggle to generalize effectively in long video understanding scenarios. To address this limitation, we adopt the YaRN context window extension technique (Peng et al., 2023; Wei & Chen, 2024), originally proposed for language models, to extend the visual context capacity of MLLMs for streaming video processing.

---

**Algorithm 1** Streaming Video Encoding and KV Cache Compression

---

**Require**: Total KV Cache size $M$, Template tokens $Q$.

  Initialize cache $K, V$ and score matrix $s$ (one row for each transformer layer).

  **while** not end of video **do**

    Fetch a new batch of frames $v_i$ from stream.

    $v_i' = \text{Filter}(v_i)$ based on frame similarity

    $K_i, V_i, s_i = \text{Encode}(v_i', Q)$

    **if** $|K| > M$ **then**

      $\mathcal{I} = \text{Topk}(s, k = M); K, V = K[\mathcal{I}], V[\mathcal{I}]$

    **end if**

    Append $K_i, V_i, s_i$ to $K, V, s$. // Equation 1

    Insert $\text{Merge}(K_i), \text{Merge}(V_i)$ to $K, V$. // Equation 2

  **end while**

---

.

| Method | Frames/FPS | KV Size | MLVU | EgoSchema | VideoMME | | |
|---|---|---|---|---|---|---|---|
| | | | | | Medium | Long | All |
| GPT-4o | - | - | 64.6 | 72.2 | 70.3 | 65.3 | 71.9 |
| MovieChat+ | 2048 | - | 25.8 | 53.5 | - | 33.4 | 38.2 |
| Dispider | 1 fps | - | 61.7 | 55.6 | 53.7 | 49.7 | 57.2 |
| LongVU | 1 fps/400 | - | 65.4 | 67.6 | 58.2 | 59.5 | 60.6 |
| LLaVA-OneVision-7B | 32 | 6K | 64.7 | 60.1 | 54.7 | 46.2 | 56.9 |
| + ReKV[†] | 0.5 fps | 353K/h | 68.5 | 60.7 | - | - | - |
| + LiveVLM | 0.5/0.2 fps | - | 66.3 | 63.0 | 56.4 | 48.8 | 57.3 |
| + StreamMem (Ours) | 0.5/0.2 fps | 6K | 66.9 | 63.0 | 56.6 | 50.1 | 59.4 |
| Qwen2-VL-7B | 768 | 50K | 65.8 | 65.2 | - | - | 63.9 |
| + Uniform Sample | - | 6K | 57.0 | 64.4 | 53.3 | 48.7 | 58.1 |
| + SnapKV | 768 | 6K | 60.7 | 62.6 | 55.3 | 51.3 | 59.6 |
| + InfiniPot-V | 768 | 6K | 65.8 | 65.6 | 60.8 | 53.4 | 62.8 |
| + StreamMem (Ours) | 4.0/0.5 fps | 6K | 65.9 | 67.2 | 62.4 | 52.3 | 62.1 |
| Qwen2.5-VL-3B | 768 | 50K | 63.3 | 64.4 | - | - | 60.3 |
| + Uniform Sample | - | 6K | 60.6 | 62.0 | 56.9 | 47.8 | 58.3 |
| + InfiniPot-V | 768 | 6K | 62.1 | 61.8 | - | - | 59.3 |
| + StreamMem (Ours) | 4.0/0.5 fps | 6K | 62.3 | 62.2 | 60.1 | 49.1 | 59.5 |

Table 1: Evaluation results of different MLLMs on offline long video understanding benchmarks. †: ReKV stores the KV cache of all seen frames so it is considered an "upper bound."

Prior works on streaming processing of long videos with MLLMs (Di et al., 2025; Ning et al., 2025; Kim et al., 2025b) reassign positional IDs to the visual tokens that are retained after KV cache compression. However, this reassignment discards the original spatial and temporal information associated with these tokens, potentially degrading performance. We demonstrate that applying YaRN with a properly chosen scaling factor (based on the MLLM's visual context window length) allows us to preserve positional consistency across streaming segments and improves performance compared to naively reassigning position embeddings.

## 5 EXPERIMENTS

### 5.1 EXPERIMENT SETUP

**Benchmarks.** We evaluate StreamMem on a number of widely used offline long video understanding benchmarks, including MLVU (Zhou et al., 2025), EgoSchema (Mangalam et al., 2023), and VideoMME (Fu et al., 2025). By default, we process the video stream at 0.5 frames per second (FPS), in accordance with ReKV Di et al. (2025). For MLVU and EgoSchema, we report the results on the official "dev" set. For VideoMME, we report results without subtitles. Each video clip is set to 8 frames. All experiments can be run with one A100 GPU.

| Method | RVS-Ego | | RVS-Movie | |
|---|---|---|---|---|
| | Acc | Score | Acc | Score |
| ReKV | 63.7 | 4.0 | 54.4 | 3.6 |
| ReKV w/o offloading. | 55.8 | 3.3 | 50.8 | 3.4 |
| Flash-VStream | 57.0 | 4.0 | 53.1 | 3.3 |
| InfiniPot-V | 57.9 | 3.5 | 51.4 | 3.5 |
| StreamMem (Ours) | 57.6 | 3.8 | 52.7 | 3.4 |

Table 2: Results of different streaming video question answering methods on RVS-Ego and RVS-Movie benchmarks.

| Method | KV Size | Holistic | S.D. | M.D. | All |
|---|---|---|---|---|---|
| Full KV | 50K | 76.3 | 73.9 | 43.3 | 65.9 |
| InfiniPot-V | 6K | 77.2 | 72.3 | 44.8 | 65.8 |
| StreamMem (Ours) | 6K | 77.5 | 72.7 | 44.4 | 65.9 |
| InfiniPot-V | 12K | 76.9 | 73.4 | 44.0 | 66.0 |
| StreamMem (Ours) | 12K | 77.7 | 73.1 | 43.9 | 66.0 |
| InfiniPot-V | 24K | 76.9 | 74.0 | 42.2 | 65.7 |
| StreamMem (Ours) | 24K | 77.6 | 73.4 | 44.5 | **66.3** |

Table 3: Comparison of InfiniPot-V and Stream-Mem on MLVU with different KV sizes. We use Qwen2VL-7B as the base MLLM.

**Models.** We apply our method on three popular open-source MLLMs: LLaVA-OneVision-7B (Li et al., 2024a), Qwen2-VL-7B (Wang et al., 2024a), and Qwen2.5-VL-3B (Bai et al., 2025).

**Baselines.** We evaluate StreamMem against strong baselines, including:

- Query-agnostic streaming video-language understanding with MLLMs, namely LiveVLM (Ning et al., 2025) and InfiniPot-V (Kim et al., 2025b), two recent streaming methods that perform KV cache compression independently of the query.
- Online video MLLMs such as MovieChat+ (Song et al., 2024) and Dispider Qian et al. (2025).
- LongVU (Shen et al., 2024), an MLLM that utilizes visual token compression for long video understanding.

We also report the performance of simple uniform frame sampling for Qwen2-7B and Qwen2.5-3B models, and ReKV Di et al. (2025) for LLaVA-OneVision, which stores the KV cache of all previously seen frames without compression. While ReKV is not feasible in memory-constrained settings for long videos, it serves as an oracle-style upper bound on performance under unbounded memory.

## 5.2 MAIN RESULTS

**Offline video understanding.** We report the main results for offline video understanding benchmarks in Table 1. For experiments with LLaVA-OneVision, we sample videos shorter than 30 minutes at 0.5 fps and videos longer than 30 minutes at 0.2 fps and constrain the GPU memory allocation below 24 GB, following the setup of LiveVLM (Ning et al., 2025). For Qwen2-VL and Qwen2.5-VL experiments, we sample video less than 3 minutes at 4.0 fps (to match the uniform sampling of 768 frames in InfiniPot-V (Kim et al., 2025b) and other videos at 0.5 fps. We keep the KV cache size at 6K per transformer layer in the MLLM.

From the results we observe that StreamMem outperforms the baselines on all benchmarks except the "long" subset of VideoMME for Qwen2-VL-7B. On LLaVA-OneVision-7B, StreamMem significantly outperforms the uniform sampling baseline with a comparable KV cache size. This highlights the benefits of streaming processing compared to uniform sampling, where significant information loss can incur in the sampling process. On Qwen2.5-VL-3B, StreamMem significantly narrows the gap between full KV and compressed KV on the challenging MLVU benchmark, showing that StreamMem also works well with smaller MLLMs, which are especially suitable for memory-constrained settings.

**Streaming video understanding.** We evaluate StreamMem on the RVS-Ego and RVS-Movie benchmarks for streaming video understanding using LLaVA-OneVision-7B (Li et al., 2024a). Unlike the offline video understanding benchmarks considered earlier, these two datasets pose open-ended question answering tasks that require models to reason over long visual contexts. Following the evaluation protocol used by prior work, we assess the generated answers using `GPT-3.5-turbo-0125`, which judges both the accuracy and an alignment score from 1 to 5. The results are provided in Table 2. Consistent with InfiniPot-V, we constrain GPU memory usage to stay below 28 GB. For ReKV without CPU offloading, it simply discards older KVs and retains only recent context as "short-term memory." The performance drop between ReKV with and with-

| Query Type | Holistic | S.D. | M.D. | All |
|---|---|---|---|---|
| True Query | 80.8 | 71.6 | 46.4 | 68.1 |
| Generic Text Query | 78.1 | 71.3 | 43.0 | 66.7 |
| Chat Template Query | 78.8 | 71.5 | 43.0 | 66.9 |

Table 4: Ablation study on different proxy queries for attention-based KV compression.

| KV Merging Strategy | Holistic | S.D. | M.D. | All |
|---|---|---|---|---|
| No Merging | 77.3 | 69.7 | 42.8 | 65.6 |
| Avg. Merging | 80.5 | 70.4 | 41.3 | 66.3 |
| Weighted Merging | 78.8 | 71.5 | 43.0 | 66.9 |

Table 5: Ablation study on the effect of different KV merging strategies.

out CPU offloading underscores the importance of maintaining long-range memory for high-quality answers.

StreamMem outperforms ReKV without offloading and is competitive with InfiniPot-V and Flash-VStream, demonstrating its effectiveness in open-ended question answering under constrained memory settings. These results highlight the method's ability to retain and utilize salient long-term information throughout streaming video.

## 5.3 Ablation Studies

We conduct ablation studies on different components in our method. For the ablation experiments, we use LLaVA-OneVision-7B (Li et al., 2024a) on the MLVU benchmark (Zhou et al., 2025). We report the average performance on the three subsets of the MLVU, namely holistic tasks (including Topic Reasoning and Anomaly Recognition), single detail (Needle QA, Ego Reasoning, and Plot QA), and multi-detail (including Action Order, and Action Count).

**Type of proxy query.** We compare the results for using different queries, including the ground truth query, the chat template query, and a generic text query ("What is happening in the video?") for the attention-based KV compression module in Table 4. We observe that the generic text query obtains similar performance to the chat template query, suggesting that the chat template query, while not including any real text, is implicitly acting as a generic query of video content. Using the ground truth user query for KV compression still significantly outperforms the query-agnostic methods, especially in "multi-detail" tasks, showing the challenge for query-agnostic methods to retain all the details required to answer the question without knowing the question during video processing.

**Merging strategy.** We compare the results for different KV merging strategies in Table 5. We observe that all frame-wise KV cache merging methods perform better than no KV merging, confirming the results from LiveVLM (Ning et al., 2025). StreamMem improved over LiveVLM which uses simple average merging and inserting to the end of each frame by applying weighted merging based on the attention scores between the chat template tokens and the visual tokens.

## 6 Conclusion

Enabling continuous video stream processing under a bounded memory constraint is essential for deploying multimodal large language models (MLLMs) in real-world, embodied scenarios. Yet, most prior work in long video-language understanding has focused on static or offline settings, assuming known queries, finite video lengths, and full access to the visual context in advance. These assumptions limit their applicability in streaming or open-world environments. In this work, we present StreamMem, a training-free and query-agnostic KV cache compression framework tailored for streaming video understanding. By using attention scores between visual tokens and chat template tokens as a proxy for query relevance, StreamMem effectively retains salient visual information without requiring access to future queries. When applied to open-source MLLMs, StreamMem achieves state-of-the-art performance across a diverse set of both offline and streaming long video benchmarks. Beyond demonstrating competitive empirical results, we conduct an in-depth analysis of various components in our framework, including input frame filtering, KV merging strategies, and positional embedding techniques, shedding light on the design considerations for constructing a memory-bounded visual processing pipeline. These insights lay a foundation for future research in scaling MLLMs to continuously process real-world visual streams.

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

## A EXPERIMENT DETAILS

**Hyper-parameter details.** In terms of the YaRN scaling factor $\lambda$, we used $\lambda = 8$ for LLaVA-OneVision, $\lambda = 2$ for Qwen2-VL, and $\lambda = 1$ (no scaling) for Qwen2.5-VL. The difference is due to the different default context length in these MLLMs. We keep the other hyperparameters the same across different models: input frame filtering similarity threshold $\delta = 0.95$; each new frame chunk has a size of 8 frames.

**Video sampling details.** Following InfiniPot-V, we use the following hyperparameter values for the Qwen2-VL vision processor: FPS_MAX_FRAMES $= 768$ and VIDEO_MAX_PIXEL$= 768 \times 28 \times 28$. The vision processor resizes each image such that the width and height are both divisible by 28; each frame is encoded into up to 130 tokens for Qwen2-VL and Qwen2.5-VL, and 196 tokens for LLaVA-OneVision.

**MLVU evaluation details.** We would like to note that there are two different ways prior papers report results on the MLVU benchmark (Zhou et al., 2025): (1) computing the overall accuracy on the entire benchmark (used by LiveVLM (Ning et al., 2025)), and (2) computing the accuracy on each task separately and average the accuracy across tasks (used by InfiniPot-V (Kim et al., 2025b)). The "overall accuracy" computed with (1) is often a bit higher than that of (2). For a fair comparison against both baselines, we use the first method for experiments using LLaVA-OneVision as the base MLLM and the second method for experiments using Qwen2VL-7B and Qwen2.5VL-3B as the base MLLM.

## B ADDITIONAL EXPERIMENTS

| YaRN Scaling Factor | Holistic | S.D. | M.D. | All |
|---|---|---|---|---|
| $\lambda = 1$ (No scaling) | 75.5 | 65.0 | 38.5 | 61.5 |
| $\lambda = 2$ | 78.6 | 69.2 | 41.9 | 65.4 |
| $\lambda = 4$ | 80.5 | 70.9 | 42.4 | 66.8 |
| $\lambda = 8$ | 78.8 | 71.5 | 43.0 | 66.9 |

Table 6: Ablation study on the effect of YaRN visual context window extension with varying scaling factors LLaVA-OneVision-7B.

| Method | KV Size | Holistic | S.D. | M.D. | All |
|---|---|---|---|---|---|
| Full KV | 50K | - | - | - | 63.3 |
| StreamMem (Ours) | 6K | 78.3 | 65.6 | 41.4 | 62.3 |
| StreamMem (Ours) | 12K | 77.6 | 67.1 | 42.6 | 63.1 |
| StreamMem (Ours) | 24K | 78.1 | 68.2 | 44.5 | 64.3 |

Table 7: Performance of StreamMem on MLVU with different KV sizes. We use Qwen2.5VL-3B as the base MLLM.

| Memory Allocation | Holistic | S.D. | M.D. | All |
|---|---|---|---|---|
| Uniform | 78.8 | 71.5 | 43.0 | **66.9** |
| Inverse Entropy | 80.3 | 70.4 | 41.3 | 66.3 |
| Exponential Decay (0.97) | 78.4 | 71.0 | 42.2 | 66.4 |
| Exponential Decay (0.98) | 80.3 | 70.1 | 42.0 | 66.2 |

Table 8: Comparison of different memory allocation strategies on MLVU with LLaVA-OneVision-7B.

**Ablation on YaRN scaling factor.** We report results for LLaVA-OneVision with different YaRN scaling factors in Table 6. We observe that YaRN visual context window extension significantly

| Input Filtering | Holistic | S.D. | M.D. | All |
|---|---|---|---|---|
| No Filtering | 77.3 | 69.7 | 42.2 | 65.4 |
| Filtering ($\delta = 0.90$) | 80.1 | 69.9 | 42.2 | 66.1 |
| Filtering ($\delta = 0.95$) | 78.8 | 71.5 | 43.0 | 66.9 |
| Filtering ($\delta = 0.97$) | 80.1 | 70.6 | 42.6 | 66.6 |

Table 9: Ablation study on the effect of frame filtering with varying thresholds.

improves the video undestanding performance, and the performance can be sensitive to different values of the YaRN scaling factor. While $\lambda = 4$ and $\lambda = 8$ give decent overall results, $\lambda = 4$ performs better on holistic tasks and $\lambda = 8$ performs better on single detail and multi-detail tasks.

**Memory Allocation Strategies.** Table 8 compares several simple memory-budget allocation strategies, including entropy-based and decay-based weighting schemes. Results show that, despite prior observations suggesting that biasing toward earlier layers can be beneficial (Chen et al., 2024b; Wang et al., 2025), a uniform allocation strategy consistently outperforms these alternatives across holistic, single-detail, and multi-detail evaluations. This suggests that uniform memory allocation remains a strong and robust baseline for managing limited memory budgets in streaming video settings.

**Input frame filtering.** We compare results for different input frame filtering thresholds in Table 9. The results confirm the benefits of input frame filtering to reduce redundancy in the input frames. In terms of the similarity threshold, we found 0.95 to be a sweet spot. While the LongVU paper (Shen et al., 2024) found that a DINO vision encoder (Oquab et al., 2023) performs better than SigLIP Zhai et al. (2023), it requires an external network which incurs additional memory and compute overhead.

# C   USAGE OF LARGE LANGUAGE MODELS (LLM)

LLMs are only used to polish the writing of this paper.

