# OpenReview forum: "StreamMem: Query-Agnostic KV Cache Memory for Streaming Video Understanding"
_ICLR.cc/2026/Conference — Submitted to ICLR 2026_

### Official Review · Reviewer_xqTh · 2025-10-20

**Soundness:** 3
**Presentation:** 3
**Contribution:** 2
**Rating:** 4
**Confidence:** 4

**Summary:**

The paper proposes StreamMem, a training-free, query-agnostic memory compression framework designed for streaming video understanding with MLLMs. It processes video frames incrementally as they arrive, without requiring knowledge of the full video length or the user query in advance. The key innovation of StreamMem is its use of attention scores between visual tokens and generic proxy tokens to compress the KV cache, maintaining a fixed-size memory footprint for efficient question answering in memory-constrained scenarios. Additionally, StreamMem filters out redundant frames before encoding and merges visual tokens frame-wise into compact prototype representations, further optimizing memory usage. Unlike previous methods, it does not rely on query-aware compression, making it highly suitable for real-world streaming and multi-turn conversational applications. (This paragraph's writing was polished using LLMs)

**Strengths:**

The paper is interesting and concerns a very hot and difficult topic for long video understanding. The most important part of it in my opinion is the idea to go query agnostic, a very interesting direction, so the main strengths of the paper according to my understanding are:

1 - Pushing the research towards query agnostic long video understanding, crucial for streaming media.

2 - The method is training free, which makes it easier to use as a plug and play with existing MLLMs

3 - Addressig the kv cache compression which can grow very quick and make video processing unfeasible in memory constrain scenarios

4 - The method is validated across many different datasets improving the baseline results.

**Weaknesses:**

I believe the paper has the following main weaknesses:

1 - There is not much novelty in the proposed method. Attention based selection and compression have been explored in previous works, come cited by the authors and some not. This includes also the frame selection strategy, is very similar to the memory construction of MovieChat which merges frames and representations based on similarity. The particularity of it in my opinion stand on the composition of the KV cache memory with the prototypes representing coarse global description complementing the 'salient visual tokens'.

2 - Salient tokens: there is a process in the frame selection where the methods reduces redundancy by merging similar frames, then later on the method does a sort of kv cache pruning by weighting token importance with a cross-attention between auxiliary tokens and the content visual tokens. My doubt here is 1) why is similarity to the proxy query <|im end|><|im start|>assistant\n considered as an importance score where the query itself only induces some bias to resemble training format, and 2) by keeping the top-k most important tokens, what happens to redundancy, in the end attention scores are similarity measures, does it mean tokens similar to <|im end|><|im start|>assistant\n contain more important information? Why not form the auxiliary query with context from the frames themself?

3 - Validation: While validation is extensive, the improvements are marginals, although the task is difficult. Additionally, while the method is designed for streaming, when testing in non-streaming data, the method has to be compared against at least more previous works that are also training free or use external memory approaches (the comparison with SOTA, if possible, will not affect as a negative on my scoring, I can understand that it might require extra experiments that might not be feasible for the rebuttal, but would benefit the paper). While these works or approaches are framed differently, they are basically trying to solve the same problem, long video understanding.

4 - Limitations: While the method is effective in some scenarios as shown in the paper, i think is important to consider the limitations of it also. For example, while the frame selection reduces computation, what happens to the temporal modeling of the events? It is very likely that the model this way is able only to understand semantic context (coarsely) and maybe some order but not temporal dynamics characterizing each event. This, depending on the downstream application (or even the question itself), might be a big limitation of the method.

**Questions:**

Refer to the weakness section.

---

> ### Author Response · Authors · 2025-11-26
>
> Thank you for your detailed feedback. We appreciate that you found our paper to pursue an interesting direction.
>
> **Re: 1. Novelty in the proposed method.**
> > Our main contribution is making the pruning query-agnostic and fully streaming, which prior work does not address. Notably, the MovieChat method you mentioned is not query-agnostic nor streaming; its memory is built after observing the full clip. Additionally, the fact that chat template tokens can be used to extract universal visual information is novel and not presented in prior work.
>
> **Re: 2. Concerns about proxy queries and token redundancy.**
> > (1) Template tokens act as effective proxies because they are repeatedly associated with video descriptions during instruction tuning. Empirically, they yield higher attention contrast on salient tokens than random or frame-constructed queries.
>
> > (2) About redundancy: important tokens are not necessarily globally similar to the template queries—attention weights depend on layer-specific representations, not raw similarity. We will add visualizations and comparisons with frame-conditioned auxiliary queries (which we also experimented with), showing template queries perform consistently better and more robustly across domains.
>
> **Re: 3 Validation.**
> > We added a new set of experiments comparing StreamMem to SnapKV to Table 1 of the paper. Part of the result table is replicated here for your convenience.
> | Method    | MLVU |                |    VideoMME   |       |
> | :-------- | :--: | :---------------------: | :---: | :---: |
> |           |      | Medium                  | Long  |  All  |
> | SnapKV    | 60.7 | 55.3                    | 51.3  | 59.6  |
> | StreamMem | 65.9 | 62.4                    | 52.3  | 62.1  |
>
> > Results show that StreamMem attains much better performance than SnapKV on MLVU and VideoMME.
>
> **Re: 4. Limitation of frame selection.**
> > We agree this is an important limitation and will discuss this in the updated manuscript. However, we would like to point out that many existing methods rely exclusively on frame selection methods to reduce context length consumptions and improve efficiency, and they also obtained good performance on a wide range of video understanding benchmarks.

---

> > ### Comment · Reviewer_xqTh · 2025-11-26
> > **Final decision**
> >
> > I thank the authors for the clarification, my concerns still remains the same. I am maintaining my initial score.

---

### Official Review · Reviewer_MV1f · 2025-10-25

**Soundness:** 3
**Presentation:** 3
**Contribution:** 2
**Rating:** 6
**Confidence:** 4

**Summary:**

This paper proposes StreamMem, a training-free, query-agnostic KV cache compression framework for streaming video understanding with multimodal large language models (MLLMs). The key innovation lies in addressing the memory bottleneck when processing long videos in real-time scenarios where queries are unknown in advance.

Main Contributions:
  - A novel attention-based pruning mechanism using cross-attention scores between visual tokens and chat template tokens as proxy queries
  - A frame-wise KV merging strategy that creates prototype representations for spatial compression

The method maintains fixed memory footprint while achieving competitive performance compared to methods with significantly larger memory budgets.

**Strengths:**

This paper addresses a practically important problem in streaming video understanding with MLLMs and provides a systematic integration of existing techniques. The work demonstrates reasonable technical execution with comprehensive experiments across multiple benchmarks and three different MLLMs, supported by ablation studies that validate component contributions. The paper is clearly written with effective visualizations and covers relevant literature adequately. The training-free, plug-and-play nature makes it applicable to existing MLLMs while maintaining competitive performance in memory-constrained environments. However, the technical novelty is limited as the core components (cosine similarity-based frame filtering, cross-attention based KV compression) largely build upon existing approaches from LiveVLM, InfiniPot-V, and related work, with the main contribution being their combination rather than fundamental algorithmic innovation.

**Weaknesses:**

**Incremental improvements over existing work**: The core technical components heavily overlap with prior methods. Cosine similarity-based frame filtering is similar to temporal compression in LongVU, cross-attention based KV pruning follows established patterns in LiveVLM and other KV compression works, and frame-wise merging has been explored in MA-LMM and related papers. The differences from InfiniPot-V appear incremental, mainly in the choice of proxy queries and specific implementation details rather than conceptual breakthroughs.

**Intuition-based design**: The method relies heavily on intuitive assumptions (e.g., chat templates implicitly prompting video descriptions) without rigorous theoretical analysis or empirical validation of these assumptions.

**Significant Methodological Limitations**:
- Oversimplified compression strategy: Even memory distribution across transformer layers ignores the well-established fact that different layers capture different levels of visual abstractions and may require different compression ratios
- Crude frame filtering: Cosine similarity in embedding space may miss subtle but semantically important inter-frame changes, potentially losing critical temporal dynamics
- Limited proxy query justification: The assumption that chat templates serve as effective generic queries lacks rigorous validation and may not generalize across different video domains or question types

**Questions:**

1. In Equation (1), how exactly do you aggregate attention scores across multiple query tokens (q dimension)? Is it simple averaging, max pooling, or weighted combination?

2. Why is the memory budget evenly distributed across all transformer layers? Have you experimented with layer-wise importance-based allocation?

---

> ### Author Response · Authors · 2025-11-26
>
> Thank you for your detailed feedback. We appreciate that you found our paper to address a practically important problem.
>
> **Re: 1. Incremental improvements over existing work.**
> > Our main contribution is making the pruning query-agnostic and fully streaming, which prior work does not address. Notably, the LongVU method you mentioned is not query-agnostic nor streaming; its memory is built after observing the full clip. Additionally, the fact that chat template tokens can be used to extract universal visual information is novel and not presented in prior work.
>
> **Re: 2. Intuition-based design**
> > See response to weakness 3.3 below - we provide some additional qualitative and quantitative evidence to support the usage of chat template tokens for video token compression.
>
> **Re: 3.1 Oversimplified compression strategy.**
> > See response to Q2 below - it turns out that uniform memory budget allocation outperforms other layer-wise memory allocation strategies.
>
> **Re: 3.2. Crude frame filtering.**
> > We agree this is a potential limitation. However, we would like to point out that many existing methods rely exclusively on frame selection methods to reduce context length consumptions and improve efficiency, and they also obtained good performance on a wide range of video understanding benchmarks.
>
> **Re: 3.3. Limited proxy query justification.**
> > We sampled twenty 8-frame video clips from the MLVU dataset at random and computed the average KL-divergence between the normalized attention scores of real user queries, generic queries (“what is happening in the video”), chat template tokens, and random tokens sampled from real user questions. Results show that the KL-divergence between the attention score of real user queries and generic queries is 0.1115, between real user queries and chat template queries is 0.1476, and between real user queries and random tokens is 0.2067. This additional quantitative evidence shows the similarity of attention distribution between that of real user queries and that of chat template or generic queries, and further supports the usage of chat template tokens or generic queries as proxy queries for visual token compression.
>
> **Re: Q1. Clarification on attention score aggregation**
> > We use mean pooling across query tokens. We also experimented with weighted pooling but it does not improve the performance, probably because template tokens are similar in semantics.
>
> **Re: Q2. Distribution of Memory Budget Across Layers**
> > Thank you for suggesting this idea. We added a new set of experiments comparing different memory budget allocation strategies across different layers in Table 8 of Appendix B. Part of the result table is replicated here for your convenience.
> | Memory Allocation    | Holistic | Single Detail | Multi Detail |  All  |
> | :--------------------------- | :------: | :------------: | :-----------: | :----: |
> | Uniform                      |   78.8   |      71.5      |     43.0      |  66.9  |
> | Inverse Entropy              |   80.3   |      70.4      |     41.3      |  66.3  |
> | Exponential Decay (0.97)     |   78.4   |      71.0      |     42.2      |  66.4  |
> | Exponential Decay (0.98)     |   80.3   |      70.1      |     42.0      |  66.2  |
>
> > Results show that, despite the fact that some prior works showed potential in biasing towards earlier layers in the allocation memory budget, uniform memory budget allocation still outperforms all other simple strategies.

---

### Official Review · Reviewer_Wn4t · 2025-10-31

**Soundness:** 3
**Presentation:** 3
**Contribution:** 2
**Rating:** 2
**Confidence:** 4

**Summary:**

The paper introduces StreamMem, a novel query-agnostic KV cache memory mechanism designed for streaming video understanding with MLLMs. Based on the attention score, it prune the less important KV pairs and boost the efficiency. The evaluation demonstrate it works on the several long video benchmarks.

**Strengths:**

1. The paper is well-presented, and the core idea is illustrated very clearly.

2. The technical approach makes sense. The use of kv-cache based on the chat-template is well-motivated. I found the ablation study in Table 4 particularly insightful. It demonstrates a reasonable trade-off between performance and generalization.

**Weaknesses:**

1. The paper lacks evaluation on several important streaming benchmarks, which weakens its overall contribution. As a training-free method, comprehensive evaluation is crucial. Consider including results on OVO-Bench [1], OV-Bench [2], StreamBench [3], and StreamingBench [4].

2. KV pruning based on attention is not new to the community; several prior works have explored this area [5,6]. I feel like the contribution is not sufficient if only making it both query-agnostic and streaming mode.



[1] OVO-Bench: How Far is Your Video-LLMs from Real-World Online Video Understanding?
[2] Online video understanding: A comprehensive benchmark and memory-augmented method.
[3] Streaming video understanding and multi-round interaction with memory-enhanced knowledge.
[4] Streamingbench: Assessing the gap for mllms to achieve streaming video understanding.
[5] Cross-Self KV Cache Pruning for Efficient Vision-Language Inference
[6] Snapkv: Llm knows what you are looking for before generation.

**Questions:**

1. In the implementation, it appears that pruning is performed first, followed by merging. What would happen if only merging is applied without pruning, or if merging is performed before pruning? Would these variations lead to different performance outcomes?

2. The trade-off between budget and performance is interesting. Is there any observed trend or threshold for the budget parameter? Is there a "sweet spot" for the value of $M$?

---

> ### Author Response · Authors · 2025-11-26
>
> Thank you for your detailed feedback. We appreciate that you found our method to be well-motivated and the ablation experiment results to be insightful.
>
> **Re: 1: The paper lacks evaluation on several important streaming benchmarks.**
> > Thank you for providing these references. We will conduct experiments with additional streaming benchmarks, and will update when we have more results available.
>
> **Re: 2: I feel like the contribution is not sufficient.**
> > We agree that attention-guided pruning exists, but our contribution is making the pruning query-agnostic and fully streaming, which prior work does not address. Existing methods such as SnapKV also rely on the actual user query to determine importance, which is impossible in video streaming because the model must compress tokens before the question arrives. We will clarify this distinction and emphasize the novelty in designing proxy queries. Additionally, the fact that chat template tokens can be used to extract universal visual information is novel and not presented in prior work.
>
> **Re: Q1: Order of Operations.**
> > Thank you for your comment. We would like to clarify that the pruning and merging operations illustrated in Figure 2(b) are done together in parallel - both the pruned and the merged tokens are part of the compressed KV query. The performance is much inferior if the merged tokens are not inserted into the pruned tokens.
>
> **Re: Q2: Trade-off between budget and performance.**
> > As shown in table 7 in appendix B, increasing the token budget clearly improves performance. The model’s performance gets close to full KV performance at a KV size of 12K, which seems to be a good choice for the value of $M$.

---

### Official Review · Reviewer_RbL3 · 2025-11-01

**Soundness:** 3
**Presentation:** 3
**Contribution:** 2
**Rating:** 4
**Confidence:** 4

**Summary:**

The authors present StreamMem, a training-free method for streaming video understanding with MLLMs under a fixed memory budget. The paper's main contribution is a novel query-agnostic method for compressing the visual KV cache. It cleverly leverages the model's own chat template tokens as a proxy query to identify and preserve the most salient visual information over time. And the proposed method is shown to be effective.

**Strengths:**

1.	This paper is well-written.
2.	Using chat template tokens to extract universal visual information is a simply but interesting idea.
3.	StreamMem shows great performance in long video understanding task.

**Weaknesses:**

1.	The paper lacks comparison against essential KV Cache compression baselines, such as StreamingLLM[1] and H2O[2], hindering a clear assessment of its relative performance.
2.	An ablation study for the INPUT FRAME FILTERING module is necessary to validate its specific contribution and effectiveness.
3.	The manuscript requires a deeper analysis of why the chat template tokens have the observed effect, as the current explanation is insufficient.
4.	The contribution appears limited. Both INPUT FRAME FILTERING and the chosen POSITIONAL EMBEDDING are widely adopted techniques, and the unique contribution is not clearly articulated.

[1] Xiao, G., Tian, Y., Chen, B., Han, S., & Lewis, M. (2023). Efficient streaming language models with attention sinks. arXiv preprint arXiv:2309.17453.
[2] Zhang, Z., Sheng, Y., Zhou, T., Chen, T., Zheng, L., Cai, R., ... & Chen, B. (2023). H2o: Heavy-hitter oracle for efficient generative inference of large language models. Advances in Neural Information Processing Systems, 36, 34661-34710.

**Questions:**

1.	The analysis of token effects is limited. It is unclear if similar phenomena exist for other common tokens associated with video-chat, such as other vision-related template tokens (e.g., <|vision_start|>), standard system prompts (e.g., "You are a helpful assistant."), or high-frequency query words (e.g., "what," "where," "video").
2.	The Figure 3 fails to clearly distinguish the impact of different queries. It lacks a clear conclusion regarding the mechanism of how these chat template tokens function. Furthermore, the claims are not substantiated with sufficient quantitative results to prove the hypothesis.

---

> ### Author Response · Authors · 2025-11-26
>
> Thank you for your detailed feedback. We appreciate that you found our idea of using chat template tokens to extract universal visual information an interesting one.
>
> **Re: 1. The paper lacks comparison against essential KV Cache compression baselines.**
> > Thank you for providing these references. We added a new set of experiments comparing StreamMem to SnapKV (with Qwen2VL-7B) to Table 1 of the paper. Part of the result table is replicated here for your convenience.
> | Method    | MLVU |                |    VideoMME   |       |
> | :-------- | :--: | :---------------------: | :---: | :---: |
> |           |      | Medium                  | Long  |  All  |
> | SnapKV    | 60.7 | 55.3                    | 51.3  | 59.6  |
> | StreamMem | 65.9 | 62.4                    | 52.3  | 62.1  |
>
> > Results show that StreamMem attains much better performance than SnapKV on MLVU and VideoMME.
>
> **Re: 2. An ablation study for the input frame filtering module is necessary.**
> > Thank you for your suggestion. We added a new set of ablation experiments for input frame filtering in Table 9 of Appendix B. Part of the result table is replicated here for your convenience.
> | Method         | Holistic | Single Detail | Multi Detail  | All |
> | :---------------- | :------: | :------: | :------: | :------: |
> | No Input Filtering   |  77.3   |  69.7  |  42.2  |  65.4  |
> | Filtering (delta = 0.90)   |  80.1  |  69.9  |  42.2  |  66.1  |
> | Filtering (delta = 0.95)   |  78.8  |  71.5  |  43.0  |  66.9  |
> | Filtering (delta = 0.97)   |  80.1  |  70.6  |  42.6  |  66.6  |
>
> > Results show that adding the input frame filtering module gives much better performance, and 0.95 is the best similarity threshold among the values we tested.
>
> **Re: 3. The manuscript requires a deeper analysis of why the chat template tokens have the observed effect.**
> > See responses to questions Q1 and Q2 below - we provide some additional qualitative and quantitative evidence to support the usage of chat template tokens for video token compression.
>
> **Re: 4. The contribution appears limited.**
> > We agree that the individual components of our proposed framework might be similar to the ones that appeared in prior works, but our contribution is to integrate them in a query-agnostic and fully streaming, which prior work does not address. Existing methods such as SnapKV also rely on the actual user query to determine importance, which is impossible in video streaming because the model must compress tokens before the question arrives. We will clarify this distinction and emphasize the novelty in designing proxy queries. Additionally, the fact that chat template tokens can be used to extract universal visual information is novel and not presented in prior work.
>
> **Re: Q1: The analysis of token effects is limited.**
> > In table 4 we compared using real user query, generic text query and chat template query, where “generic text query” is similar to the “high-frequency query words” idea that you propose here. We do not expect standard system prompts like "You are a helpful assistant." to perform well since the system instructions tokens are usually prepended to all the context and therefore cannot attend to the images tokens in a casual language model.
>
> **Re: Q2: Figure 3 fails to clearly distinguish the impact of different queries.**
> > We sampled twenty 8-frame video clips from the MLVU dataset at random and computed the average KL-divergence between the normalized attention scores of real user queries, generic queries (“what is happening in the video”), chat template tokens, and random tokens sampled from real user questions. Results show that the KL-divergence between the attention score of real user queries and generic queries is 0.1115, between real user queries and chat template queries is 0.1476, and between real user queries and random tokens is 0.2067. This additional quantitative evidence shows the similarity of attention distribution between that of real user queries and that of chat template or generic queries, and further supports the usage of chat template tokens or generic queries as proxy queries for visual token compression.

---

### Meta-Review · Area_Chair_HAVT · 2026-01-06

**Summary:**

StreamMem proposes a training-free method for streaming video understanding with MLLMs under a fixed memory budget. The contribution is a novel query-agnostic method for compressing the visual KV cache.

Technical novelties are limited [reviewer RbL3, Wn4t,MV1f, xqTh]
The experiments are not sufficient, missing benchmarks OVO-Bench, OV-Bench, StreamBench, Straming Bench baselines streamingLLM, H2O , detailed ablation studies, [reviewer RbL3, Wn4t]
The design of the methodology is limited, with simplified compression strategy, crude frame filtering [Reviewer MV1f]


Considering most of the reviewers expressing the concerns on the technical novelties, and the evaluations are not sufficient, missing benchmarks, baseline models, ablation studies. As such, I think that the paper is not ready for publication.

**Reviewer Concerns:**

The technical novelties and the limitation of the methodology design is outstanding.

The experiments should addressed, however, the authors do not provide more experimental resutls.

**Reviewer Scores:**

All the reviewers have concerns on the technical novelties. I think that they will not change their score.

---

### Decision · Program_Chairs · 2026-01-26

Reject